# Cell-free biosynthesis combined with deep learning accelerates de novo-development of antimicrobial peptides

Amir Pandi [1] ✉, David Adam[1,2], Amir Zare [1], Van Tuan Trinh [3], Stefan L. Schaefer [4], Marie Burt[5], Björn Klabunde[5], Elizaveta Bobkova[1], Manish Kushwaha [6], Yeganeh Foroughijabbari[1], Peter Braun [2,7,8], Christoph Spahn [9], Christian Preußer[10,11], Elke Pogge von Strandmann[10,11], Helge B. Bode[9,12,13,14,15], Heiner von Buttlar[2,7], Wilhelm Bertrams [5], Anna Lena Jung[5,16], Frank Abendroth[3], Bernd Schmeck [5,15,16,17,18,19], Gerhard Hummer [4,20], Olalla Vázquez [3,15] & Tobias J. Erb [1,15] ✉

Bioactive peptides are key molecules in health and medicine. Deep learning holds a big promise for the discovery and design of bioactive peptides. Yet, suitable experimental approaches are required to validate candidates in high throughput and at low cost. Here, we established a cell-free protein synthesis (CFPS) pipeline for the rapid and inexpensive production of antimicrobial peptides (AMPs) directly from DNA templates. To validate our platform, we used deep learning to design thousands of AMPs de novo. Using computational methods, we prioritized 500 candidates that we produced and screened with our CFPS pipeline. We identified 30 functional AMPs, which we characterized further through molecular dynamics simulations, antimicrobial activity and toxicity. Notably, six de novo-AMPs feature broad-spectrum activity against multidrug-resistant pathogens and do not develop bacterial resistance. Our work demonstrates the potential of CFPS for high throughput and low-cost production and testing of bioactive peptides within less than 24 h.

According to the world health organization, antimicrobial resistance (AMR) is among the top 10 global health threats[1]. In 2019 alone, multidrug-resistant bacteria including pathogenic *Escherichia coli*, ESKAPE pathogens (*Enterococcus faecium, Staphylococcus aureus, Klebsiella pneumoniae, Acinetobacter baumannii, Pseudomonas aeruginosa, Enterobacter* spp.), *Streptococcus pneumoniae*, and *Mycobacterium tuberculosis* caused 1.27 million deaths[2]. This number is predicted to reach 10 million annually by 2050[2]. Despite this looming threat, the development of new antimicrobials is lagging behind. While >4000 immuno-oncology compounds were in clinical trials in 2021, only 40 antimicrobials (of which none is active against multi-drug resistant Gram-negative bacteria) were subjected to clinical studies[3], highlighting the urgent need to increase the development of novel antimicrobial compounds.

One promising class of antimicrobial compounds are antimicrobial peptides (AMPs)[4–8]. A big class of AMPs contains linear peptides of 12–50 canonical amino acids (AA), which have evolved as part of nature's antimicrobial arsenal of bacteria as well as the innate immune system of multicellular organisms[4,6,8]. Compared to classical antibiotics, AMPs show decreased resistance development mainly because (i) most AMPs act directly at the cell membrane, (ii) show a relatively high killing rate, and (iii), resistance against AMPs is conferred by rather non-specific mechanisms, which reduces the chances of mutational and/or horizontal gene transfer events[4]. Overall, this makes AMPs interesting candidates for next-generation antimicrobials.

About 5000 AMPs have been characterized to date, most of which are of natural origin. However, these 5000 AMPs span only a tiny

fraction of the possible solution space that nature could have explored ($\sim 20^{30}$ for a 30 AA AMP). Additionally, AMP mining from genomes and metagenomes is hampered by limited natural (and not discovered yet) AMPs as well as by available computational and experimental AMP mining tools. Hence, our ability to discover new AMPs from this *terra incognita* is limited. As randomly generated peptides are less likely to show antimicrobial properties[9], a way of unlimited development of AMPs is the use of deep learning models, which are increasingly employed for de novo protein and peptide design[10–14]. In this approach, known as generative deep learning, models learn the natural protein sequence landscape in training sets with unlabeled data to propose new-to-nature protein sequences[15]. These models are distinct from predictive models that use labeled data to predict specific properties (labels) of proteins from their sequence[16]. Generative and predictive deep learning have been recently used for the discovery of novel AMP sequences, which have been subsequently created and validated through chemical synthesis of the individual candidates[6,17–20]. While this proof-of-principle showcased the potential of deep learning in AMP discovery, a broader application of this approach has been limited due to the lack of convenient methods for the production and screening of more AMP candidates in medium to high-throughput.

One possibility to increase the throughput in AMP production is to switch from chemical synthesis to DNA-based bioproduction methods. However, heterologous expression of AMPs in microorganisms, such as *E. coli*, features several disadvantages: (i) it is time- and labor-intensive, (ii) it requires the cloning, production and purification of AMPs from cell cultures, and most importantly, (iii) many (potent) AMP candidates might not be available, as they potentially kill the producer strain upon induction. Cell-free protein synthesis (CFPS) offers a promising solution to these challenges. CFPS systems are in vitro transcription translation (TX-TL) systems that directly use DNA templates for protein biosynthesis[21–23], which allow the production of peptides outside of living cells. Thus these systems can help overcome potential cellular toxicity effects, and open up the way for the rapid, small-scale production of several hundreds of peptides from linear DNA in parallel.

Here, we combined deep learning and CFPS for de novo-design, rapid production and screening of AMPs at small scale within 24 h, and <10$ per individual AMP production assay (excluding cost for the DNA fragment). Having explored ~500,000 theoretical sequences, we screened 500 AMP candidates to identify 30 functional AMPs, which are completely unrelated to any natural sequences. Notably, six of these AMPs exhibited high antimicrobial activity against multidrug-resistant pathogens, showed no emergence of resistance and only minimal toxicity on human cells.

## Results

### De novo AMP design using deep learning

For de novo-design of AMPs, we adapted two versions of deep generative variational autoencoders (VAE) from previous studies[20,24] different in their loss function (Methods). Generative VAE are unsupervised learning models, which take as input only AMP sequences and comprise an encoder, a latent space, and a decoder. During model training, the encoder compresses the input sequences into a low-dimensional space (latent space), while the decoder aims at reconstructing sequences from this latent space (Fig. 1a). We first pretrained the VAE using ~1.5 million peptide sequences from UniProt as a generic dataset. Second, performed transfer learning on the pretrained VAE using a dataset of ~5000 known AMPs to set up the latent space to be used for de novo AMP generation (Methods, Supplementary Table 1).

To reduce the number of AMPs for experimental testing, we set up a method to select potential candidates according to their predicted bioactivity, i.e., their Minimum Inhibitory Concentration (MIC). To that end, adapted from previous works or built in-house (Methods), we established predictive deep learning models that we trained with the sequence and respective experimental MIC values of ~5000 known AMPs (sequence-MIC relationship, Fig. 1b). As regressors, we used convolutional neural networks (CNN) and recurrent neural networks (RNN) (Methods, Supplementary Tables 1-3).

To identify interesting AMP candidates, we first generated new AMPs by sampling points from the latent space and subsequently

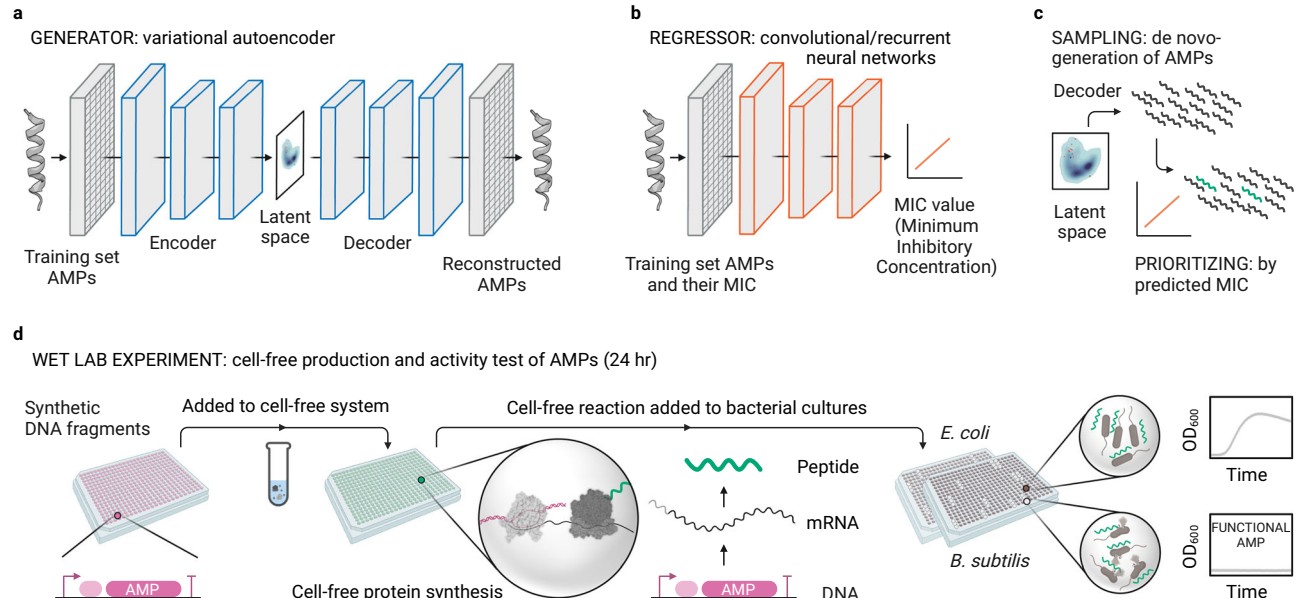

**Fig. 1 | The workflow for de novo-development of AMPs via deep learning and cell-free biosynthesis. a** Generative variational autoencoders (VAE) for de novo-design of AMPs after being trained on known AMP sequences. **b** Predictive convolutional or recurrent neural networks as regressors for the MIC prediction after being trained on known AMPs and their MIC. **c** Trained generative and predictive models are used for sampling from the latent space (de novo-design of AMPs) and prioritization of AMPs (predicting their MIC), respectively. **d** Experimental pipeline for rapid cell-free biosynthesis of the designed AMPs from synthetic DNA fragments and direct testing of produced AMPs in the cell-free mix to bacterial cultures followed by overnight continuous growth assay. Created with BioRender.com.

feeding them into the decoder, which yielded peptide sequences that share the same properties but are novel compared to the training dataset. These de novo AMPs were then prioritized by the regressors according to their predicted MICs. In five rounds, using different versions of models, we generated ~500,000 new peptides by sampling from the latent space. We filtered these peptides by length and viability to ~50,000 candidates and prioritized 500 AMP candidates for wet lab bioactivity test (Fig. 1c and Supplementary Table 4).

### Cell-free biosynthesis enables rapid screening for functional AMPs

To establish CFPS-based screening of AMPs, we designed an experimental pipeline for the high-throughput synthesis and testing of AMPs in 384-well format (Fig. 1d). The system is based on linear DNA templates, which comprise a T7 promoter and a ribosome binding site (RBS), to initiate transcription (TX) and translation (TL), followed by the AMP coding region, and a T7 terminator. After adding the DNA template (10 nM) directly into 10 µL of a cell-free TX-TL system, AMPs were produced within 4 h (Methods). To test the antimicrobial activity of the in vitro-produced peptides, 4 µL of the cell-free mix was added into a final volume of 20 µL cultures of *E. coli* (Gram-negative) and *Bacillus subtilis* (Gram-positive). Following $OD_{600}$ measurement for 20 h allowed identification of those peptides that show antimicrobial activity by suppressing growth. Overall, the entire process of CFPS with subsequent bioactivity tests takes ~24 h, as the system works with linear DNA and does not require any extensive cloning or peptide purification steps.

First, we validated the screening pipeline with two known AMPs, BP100[25] and Cecropin B[26], and then screened 500 AMP candidates (see above) in five subsequent design-predict-build-test cycles to identify 30 functional de novo AMPs (Fig. 2, Supplementary Table 5). During these five rounds, the success of functional AMP discovery increased from 0% to 12.7% from the first to the fifth round, respectively (Supplementary Table 4). Because translation initiation rates (TIRs) strongly affect protein yield[27] we tested whether our screen had biased against candidates with low TIR. We calculated the TIR for all sequences tested[28], but could not find a significant difference between the 500 candidates tested and the 30 functional AMPs identified (Supplementary Note 1, Supplementary Fig. 3). Functional AMPs were re-validated in biological triplicates with 10 µL of cell-free mix added to the final volume of 20 µL cultures (Fig. 2b, Supplementary Fig. 1) and production of AMPs was analyzed through SDS-PAGE (Supplementary Fig. 2).

### Functional AMPs are unique but share properties with natural counterparts

We next analyzed our de novo-designed AMPs in more detail. AlphaFold[29] predicted that 27 out of the 30 sequences form a helical structure (Fig. 2b), which is a common feature of many AMPs[4,5]. Interestingly, AMPs #1-19, generated by VAE 1, showed more structural diversity compared to AMPs #20-30, generated by VAE 2, which is in line with the fact that the two VAEs create two different latent spaces (Fig. 2c), thus generating AMPs with distinct structural, physico-chemical, and sequence features in a robust manner (Supplementary Fig. 4). One of the main characteristics of AMPs is an amphiphilic character that results from alternating cationic and hydrophobic amino acids in the AMP core[4,5]. In contrast to natural AMPs, which mainly feature aliphatic amino acids, the hydrophobic core of our de novo AMPs was mostly aromatic (Supplementary Fig. 5a). Phenylalanine was particularly overrepresented at the cost of leucine, which was underrepresented (Supplementary Fig. 5b, Supplementary Table 6). While there are significant physicochemical differences between training AMPs and generated peptides as well as between the generated and prioritized peptides, the 500 tested and 30 functional AMPs did not show such differences probably because of either the small

sample size or the differences that separate functional candidates from the rest cannot be shown by simple physicochemical indices. Additionally, BLAST searching showed that our de novo AMPs were unique in their sequence. No significant similarity was observed against the UniProt database, encompassing ~240 million entries, nor an AMP sequence from the training dataset. (see Supplementary Tables 7-9, and Supplementary Note 2 for detailed BLAST sequence similarity analyses). Altogether, these results demonstrated that our de novo AMPs shared the physico-chemical building principles with their natural counterparts, but were distinct from them in their amino acid sequences.

### De novo AMPs prefer bacterial over human membranes

Structural and sequence analysis suggested that our de novo AMPs act as amphipathic helices that insert into membranes. We used molecular dynamics (MD) simulations to study the interaction of our AMPs with models of a negatively charged inner membrane of bacteria (IM) and the human plasma membrane (PM) (Fig. 3a, Supplementary Note 3). According to our simulations, all AMPs bind much stronger to the IM interface than to the PM (Fig. 3b) primarily due to the interaction of such cationic AMPs with negatively charged bacterial membranes. Binding of the AMPs at the IM progressed rapidly, taking at most 200 ns to fully insert into the membrane interface (Supplementary Fig. 6a). Once bound, AMPs stayed tightly bound to the IM for the remainder of the simulations. In some cases, we observed a reorientation of the AMP after a few hundred nanoseconds from a shallowly bound state to a binding mode that resided deeper in the membrane (e.g., AMP #29). Several AMPs also partially bind the PM. However, in most cases, this binding is transient with frequent un- and rebinding and without penetrating deeper into the PM, as seen for the IM (Supplementary Fig. 6b). Furthermore, all AMPs show a higher number of (mainly electrostatic) interactions with the IM than with the PM (Supplementary Fig. 6c). While the predicted (mostly helical) structure of most of the AMPs is largely preserved in our simulations, the stronger membrane binding to the IM goes hand in hand with increased stability of the secondary structures when compared with the often fully solvated AMPs in the PM systems (Supplementary Fig. 6d). The MD simulations suggested that all our AMPs generally target bacterial membranes over the human plasma membrane, naturally however, the degree of preference is dependent on the individual AMP.

### De novo AMPs show favorable MIC to toxicity ratios

Because the concentration of peptides in CFPS is not defined, we needed to have pure peptides for cellular assays. To obtain pure compounds, we chemically synthesized the functional AMPs and characterized their bioactivity, in particular the minimum inhibitory concentration (MIC)[30], as well as hemolysis (HC50) and cytotoxicity (CC50), both expressed as 50% toxic concentration. Of the 30 candidates, 22 peptides were successfully produced by chemical synthesis. Twenty AMPs showed a MIC of ≤6 µM on *B. subtilis*, and fifteen AMPs showed a MIC of ≤25 µM on *E. coli* (Fig. 4a, Supplementary Tables 10, 11). HC50 and CC50 were significantly higher in most cases, with fifteen AMPs showing HC50 > 100 µM (thirteen > 250 µM) against fresh human red blood cells, and thirteen AMPs showing CC50 > 100 µM (six > 250 µM) against HCT116 human colon cells (Fig. 4a, Supplementary Table 10, 11). This indicates that the bioactivity versus toxicity relationship was very favorable for several of our de novo AMPs. We decided to continue with sixteen AMPs that showed a favorable bioactivity to toxicity ratio and excluded six AMPs because of high MIC and/or low HC50/CC50 values (AMP #1, #7, #10, #18, #23, and #26).

### De novo AMPs show broad-band activities in vitro

In the following, we tested the sixteen remaining de novo AMPs against clinically relevant strains, and in particular multidrug-resistant ESKAPE

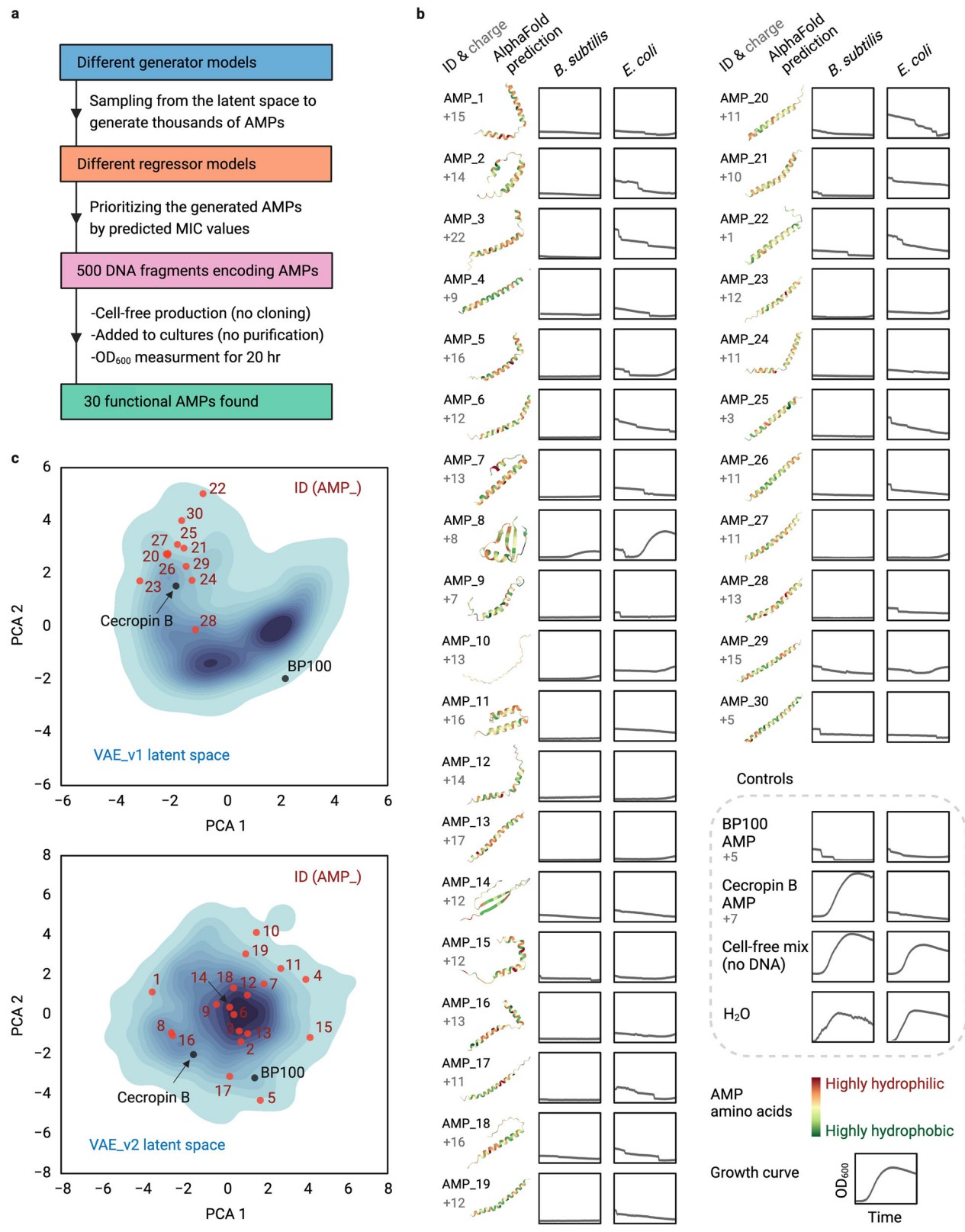

pathogens (i.e. *Enterococcus faecium, Staphylococcus aureus, Klebsiella pneumoniae, Acinetobacter baumannii, Pseudomonas aeruginosa,* and *Enterobacter* spp.; Fig. 4b). Our AMPs were most potent against *E. faecium* and *A. baumannii*, resulting in fifteen AMPs with MIC ≤ 25 μM (thirteen ≤6.3 μM), and thirteen AMPs with MIC ≤ 25 μM (ten ≤ 6.3 μM), respectively (Fig. 4b). For the rest of the ESKAPE pathogens, *K. pneumoniae, P. aeruginosa, S. aureus,* and *Enterobacter* spp., ten, nine, eight,

and three AMPs showed MIC ≤ 25 μM, respectively (Fig. 4b). While some AMPs showed distinct activity profiles against individual strains, six AMPs classified as broad-spectrum antimicrobials that showed favorable therapeutic window i.e., antimicrobial activity at relevantly low hemolysis and cytotoxicity (AMP #3, #5, #13, #15, #16, and #27). Notably, these AMPs were also active against the notorious biothreat agents *Yersinia pestis* and *Bacillus anthracis* (Fig. 4b).

**Fig. 2 | Cell-free production of de novo-generated and prioritized AMPs and activity screening against *B. subtilis* and *E. coli*. a** We used different generative and regressor models (Supplementary Table 4) to design and prioritize AMPs in five rounds, produced and screened a total number of 500 AMPs from synthetic DNA fragments and found 30 functional candidates. **b** Charge and AlphaFold-predicted structure of the functional AMPs with associated slowed/stopped growth curves for *B. subtilis* and *E. coli*. All (including control) AMPs were produced using CFPS and no peptide purification was carried out prior to the activity test. Growth curves (OD$_{600}$ 0–0.45 over time 4–20 h for all) are the average of *n* = 3 independent experiments.

Growth curves with error bars as standard deviation are provided in Supplementary Fig. 1. **c** 2D-projections of the 50-dimensional latent space were obtained by principal component analysis (PCA) for the two variational autoencoders (VAEs, without and with the KL-term annealing, VAE_v1 and VAE_v2, respectively, see Methods) that were used for de novo-design of AMPs. Blue color intensity represents the frequency of training AMPs in the latent space. Functional AMPs (red), BP100 and Cecropin B (black) annotated back into the latent space. Source data for b are provided as a Source Data file. Created with BioRender.com.

## No resistance was developed against de novo AMPs in vitro

Next, we tested the emergence of resistance against the six broad-band AMPs identified in this study (AMP #3, #5, #13, #15, #16, and #27). To that end, we performed *E. coli* serial passaging experiments with the peptides. As control, we added imipenem, a broad-spectrum antibiotic that is generally considered a last resort against multidrug-resistant pathogens[31]. During 21 days of serial passage, we did not observe a significant increase in MIC of our AMPs, while the imipenem MIC gradually increased up to 8-fold and exceeded the susceptibility breakpoint of clinically relevant resistance defined by EUCAST (Fig. 5a, Supplementary Fig. 7). The fact that we did not observe resistance is in line with the fact that our AMPs do not act on a specific cellular target, but rather globally (i.e., at the membrane, Fig. 3c), which makes them less likely to cause resistance development. This mode of action was further confirmed by propidium iodide staining and microscopy (Fig. 5c, d), which clearly indicated membrane disruption upon AMP treatment.

Finally, we also investigated the effect of AMP treatment on the release of outer membrane vesicles (OMVs) in *E. coli*. OMVs are naturally released by Gram-negative bacteria and can be a reaction against surface attacking agents neutralizing the effect of membrane targeting antibiotics[32–34]. Notably, none of the six broad-band AMPs did significantly increase the release of OMV compared to untreated *E. coli*, unlike polymyxin B, an antibiotic acting on the outer membrane of Gram-negative bacteria (Fig. 5b). Altogether, these experiments suggested that our six broad-band AMPs are able to escape resistance development and self-defense reactions of bacteria.

## Discussion

In this work, we describe the design and validation of 30 de novo AMPs, of which six show broad-band activity in vitro. Recent work has impressively demonstrated the power of deep learning methods in discovering and/or developing novel AMPs[6,20]. However, these efforts still suffer from the limited number of peptides that can be synthesized and tested, and the relatively long time it takes from design to validation. Here we used CFPS to dramatically advance the design-build-test cycle in AMP development.

Although in this study we employed CFPS pipeline to linear peptides with natural amino acids, a unique advantage of CFPS that can be leveraged in the future is the possibility to synthesize peptides containing cycles and noncanonical amino acids. Similar to chemical synthesis which has limitations such as our experience with 8 out of 30 AMPs that did not result in successful chemical synthesis, CFPS can face difficulties in expressing peptides with difficult-to-synthesize motifs. Additionally, peptide expressibility and structuring, as well as possible interactions with the PURE components can lead to false negative results. Fortunately, the throughput of CFPS compensates for such cases by screening a high number of candidates. Note that the longer the peptide sequences (higher antimicrobial specificity), the more difficult chemical synthesis and less difficult CFPS work. Additionally, our CFPS pipeline can be applied to big proteins.

Compared to recent approaches, our success rate in finding functional AMPs is in the same range, 6% for all candidates tested (12.6% for our best generator-regressor combination, Supplementary Table 4) versus 10% reported by Das et al.[20]. However, the total number

of functional AMPs discovered is an order of magnitude higher (30 AMPs versus 2) and at massively increased rate (24 h versus 28 days) and reasonably low cost (<10$ for production of one AMP for screening on two strains in parallel, excluding the ever-decreasing cost of DNA synthesis or alternatively using PCR primers). Although, state-of-the-art deep learning models can improve the hit rate in future works, high throughput approaches such as ours enable achieving higher hit numbers. Additionally, in-house preparation of CFPS systems could reduce the cost of peptide production down to ~1$[35,36].

While being unique and diverse, our de novo AMPs share common properties with known AMPs. They are predicted to be mostly α-helical peptides rich in cationic and hydrophobic amino acids and preferably act on negatively-charged IMs, showing that our pipeline was able to design new-to-nature sequences that follow the general building principles of AMPs. The resulting AMPs have several features that (after in vivo validation) could contribute to their successful translation into therapeutic applications, including broad-spectrum activity, a low propensity for bacterial resistance development, potential for topical or systemic use, and synergistic potential with existing therapies.

Although our primary focus was on AMPs with broad-spectrum activity, we note that our pipeline, combined with various machine learning techniques, is also well-suited for the development and iterative optimization of AMPs with more specific characteristics. These features may include selectivity and specificity, stability, in vivo bioavailability, immunomodulatory properties, synergy with existing drugs, and resistance. By refining these features, our pipeline has the potential to advance the design of AMPs for a variety of clinical applications.

Overall, our work provides a proof-of-principle, how CFPS can be used to leverage the full potential of machine learning approaches in the future. Especially in the light of ever-decreasing DNA synthesis costs, our combined approach of deep learning and CFPS provides a time-, cost-, and labor-effective approach for peptide production and screening. Thus, our work holds the potential to explore the design-function space of AMPs at increased rate and depth. This will hopefully lead to the increased discovery and development of peptide-based drug candidates in the future.

## Methods

### Pretraining and training datasets

Pretraining data. To gather a large corpus of protein sequences representing a general protein grammar, we downloaded all protein sequences shorter than 49 amino acids from UniProt[37] (as of July 2021). After removing duplicate sequences and entries with unknown amino acid characters, 3,104,952 unique sequences remained of which a random subset of half of them was used for pretraining.

AMP data. For experimentally validated AMP sequences, we used the Giant Repository of AMP Activities (GRAMPA)[38] which has combined sequence and activity data from several public AMP databases; APD[39], DADP[40], DBAASP[41], DRAMP[42], and YADAMP[43]. This database consists of 6,760 unique sequences and 51,345 total MIC measurements, spanning several bacterial and nonbacterial target species. We filtered the MIC measurements to the ten most abundant bacterial

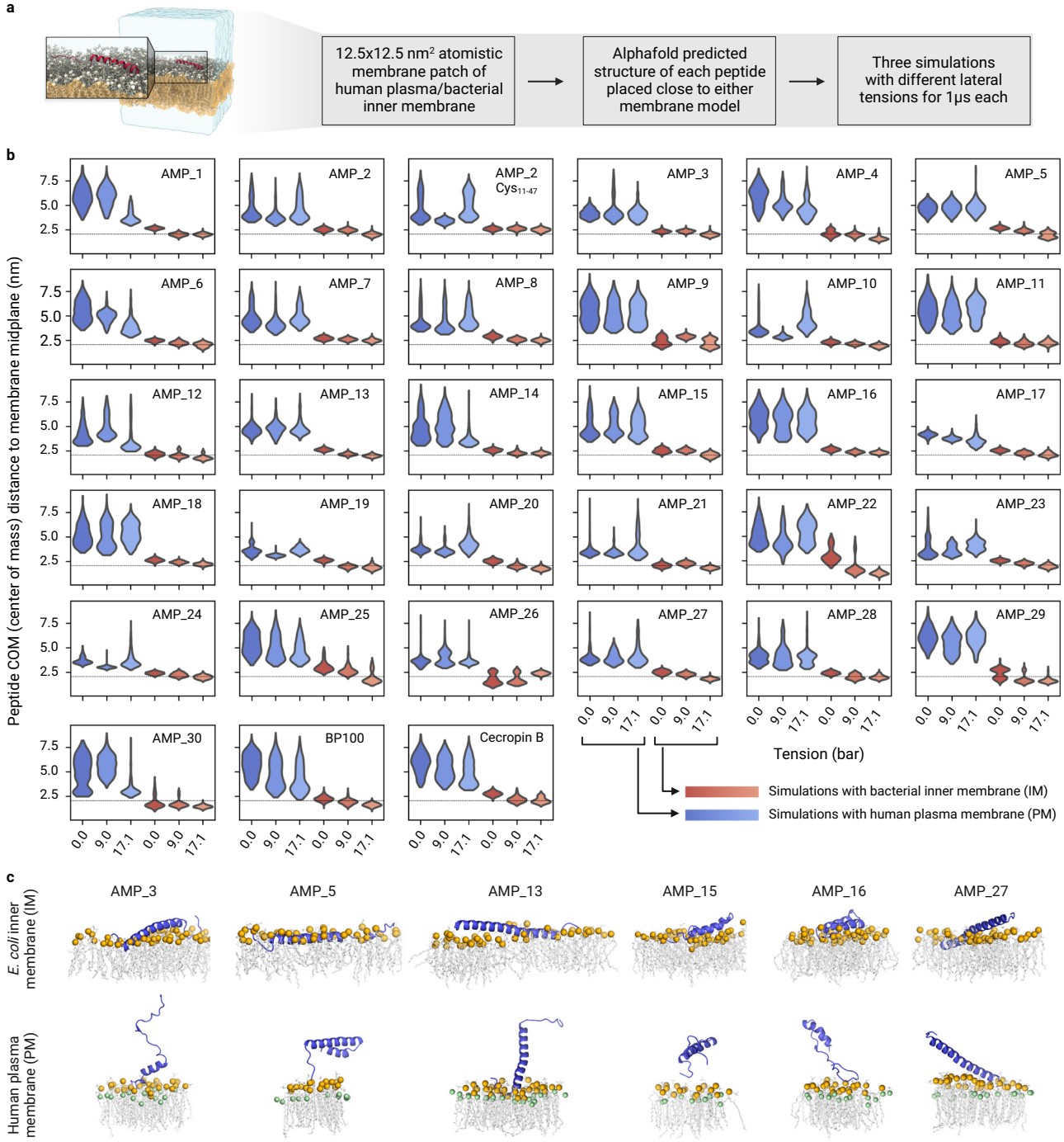

**Fig. 3 | Molecular dynamics (MD) simulations of AMP interactions with membranes. a** Overview of simulation setup and exemplary snapshot of AMP #5 after 1 μs of simulated time on the IM. The peptide is shown in red cartoon representation, solvent (water + ions) is shown as transparent blue surface and the membrane is shown as gray licorice representation with lipid headgroup phosphates shown as spheres. **b** Distributions of distances between the centers of mass of the AMPs and the membrane midplane along the direction of the membrane normal (y-axis). Distributions are calculated from the last 940 ns of 1 μs long replicates, run with different lateral membrane tensions (x-axis) and with different membranes (blue: PM; red: IM). The thin dotted line indicates the headgroup phosphate positions. AMP #2 was simulated with and without disulfide bond (Methods). **c** Rendered simulation snapshots of potent AMPs (Fig. 4) on IM (top) and PM (bottom). See Fig. 5c, d for the mode of action experiments. The interacting leaflet of the membrane is depicted with gray lipid tails and orange (phosphates) and green (cholesterol oxygen) spheres. Created with BioRender.com.

species (*E. coli, P. aeruginosa, Salmonella typhimurium, K. pneumoniae, A. baumannii, S. aureus, B. subtilis, S. epidermidis, Micrococcus luteus and E. faecalis*) and omitted all peptides containing any chemical modification other than C-terminal amidation, for feasible in-laboratory expression. After removing duplicate sequences and entries >48 amino acids, 5319 unique AMP sequences were left.

**Non-AMP data.** We searched the UniProtKB[37,44] for proteins labeled as "NOT antimicrobial, antibiotic, antiviral or antifungal" (downloaded as of July 2021), removed entries containing ambiguous amino acids, and kept only unique sequences shorter than 49 amino acids. This resulted in a dataset containing 10,612 unique non-AMP sequences.

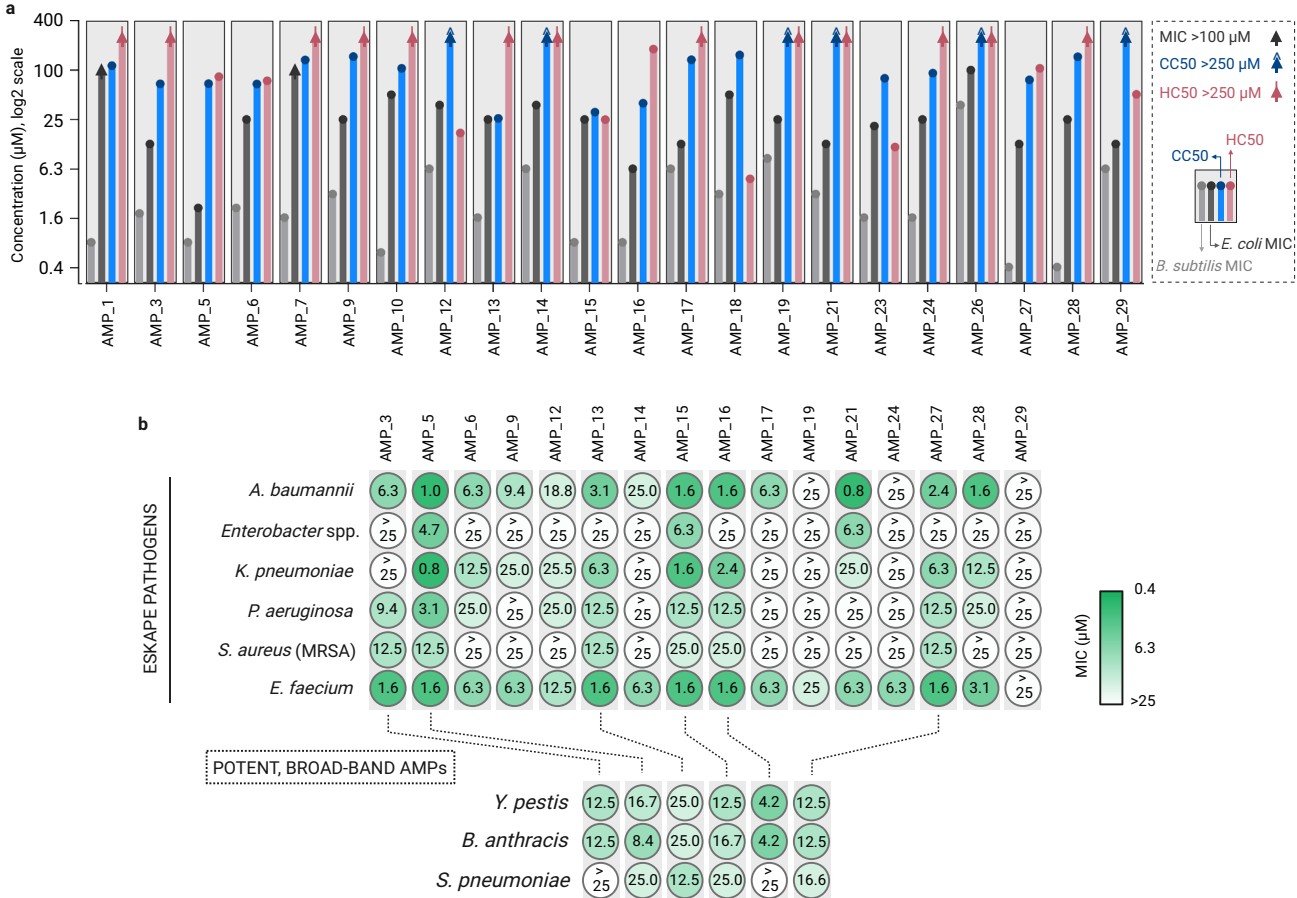

**Fig. 4 | Bioactivity characterization of chemically synthesized functional AMPs.**
**a** Minimum inhibitory concentration (MIC) of the AMPs against *B. subtilis* (gray) and *E. coli* (black), average of *n* = 3 independent experiment. CC50 (50% cytotoxicity, blue) and HC50 (50% hemolysis, red) values of the AMPs on HCT116 human colon cells and human red blood cells, respectively (average of *n* = 2 independent experiments). Arrows show values above the measured limits, black, blue and red for MIC > 100 μM, CC 50 > 250 μM and HC50 > 250 μM, respectively. MIC, HC50 and CC50 values are provided in Supplementary Table 10. **b** MIC values of the AMPs tested against ESKAPE pathogens including *E. faecium*, Methicillin-resistant *S. aureus* (MRSA), *K. pneumoniae*, *A. baumannii*, *P. aeruginosa*, *Enterobacter* spp measured in duplicates (*n* = 2 independent experiments) and MIC of the six potent, broad-band AMPs on *Y. pestis* and *B. anthracis* and *S. pneumoniae* as the average of triplicates (*n* = 3 independent experiments). Created with BioRender.com.

## Generator variational autoencoder (VAE)

We used VAEs because they have previously been used for de novo AMP design[17,18,20,45]. The generative VAE consists of an encoder, a latent vector, and a decoder. The encoder feeds the input data (one-hot encoded amino acid letter of peptides) into a latent vector that is an information bottleneck, and the decoder aims to reconstruct the input data from the latent vector. Training the VAE and minimizing the difference between the input data and the reconstructed data acts two-fold; the encoder learns to map the training dataset into a lower-dimensional space and the decoder learns to generate samples similar to the training data from any vector in the latent space. Thus, each peptide in the training dataset lands on a point in the multi-dimensional latent space. Picking vectors from the empty regions in this space and feeding them into the decoder yield peptide sequences that share the same grammar but are novel and not seen in the training dataset. After pretraining and training (transfer learning), we generated new AMPs by sampling from the latent space using different strategies in particular by exploring the neighborhood of a control functional AMP, gradient descent, or random sampling (Supplementary Tables 1, 4).

Generator VAE models. We adapted the neural network architecture of the CNN-RNN hybrid VAE model from Hawkins-Hooker et al.[24]. The encoder consists of 5 consecutive one-dimensional convolution layers fed into a dense layer of size 50, which is the latent vector. The decoder is made from 4 deconvolution layers that samples the latent vector and a GRU layer of 512 cells outputs a sequence in the same dimension as the input. The model loss is the weighted sum of a reconstruction loss and the Kullback-Leibner (KL) loss. The total loss function can be dynamically changed in the training process for KL-term annealing[46]; as standard practice when working with discrete data such as language. Our two final models were trained without and with the KL-term annealing (VAE_v1 and VAE_v2, respectively). The models were compiled using the Adam optimizer.

Pretraining and training. Our training dataset of ~5000 AMP sequences is not sufficient for learning what makes a protein sequence distinct from a random string of amino acid characters and what makes a protein sequence an AMP. In such cases, pretraining with a much bigger generic dataset is needed to enhance the model performance. We pretrained the generator models with ~1.5 million protein sequences from UniProtKB for 600 epochs. We then trained the models on the AMP data for 400 epochs. Model training metrics are provided in Supplementary Table 1.

## Regressor convolutional and recurrent neural networks

Regressor neural networks. We adapted a regressor model previously reported[38]. First, a CNN regressor was built with two consecutive one-dimensional convolutional layers, a max pooling layer, a flattening layer, a dropout layer (0.5), and three dense layers. Second, we built a

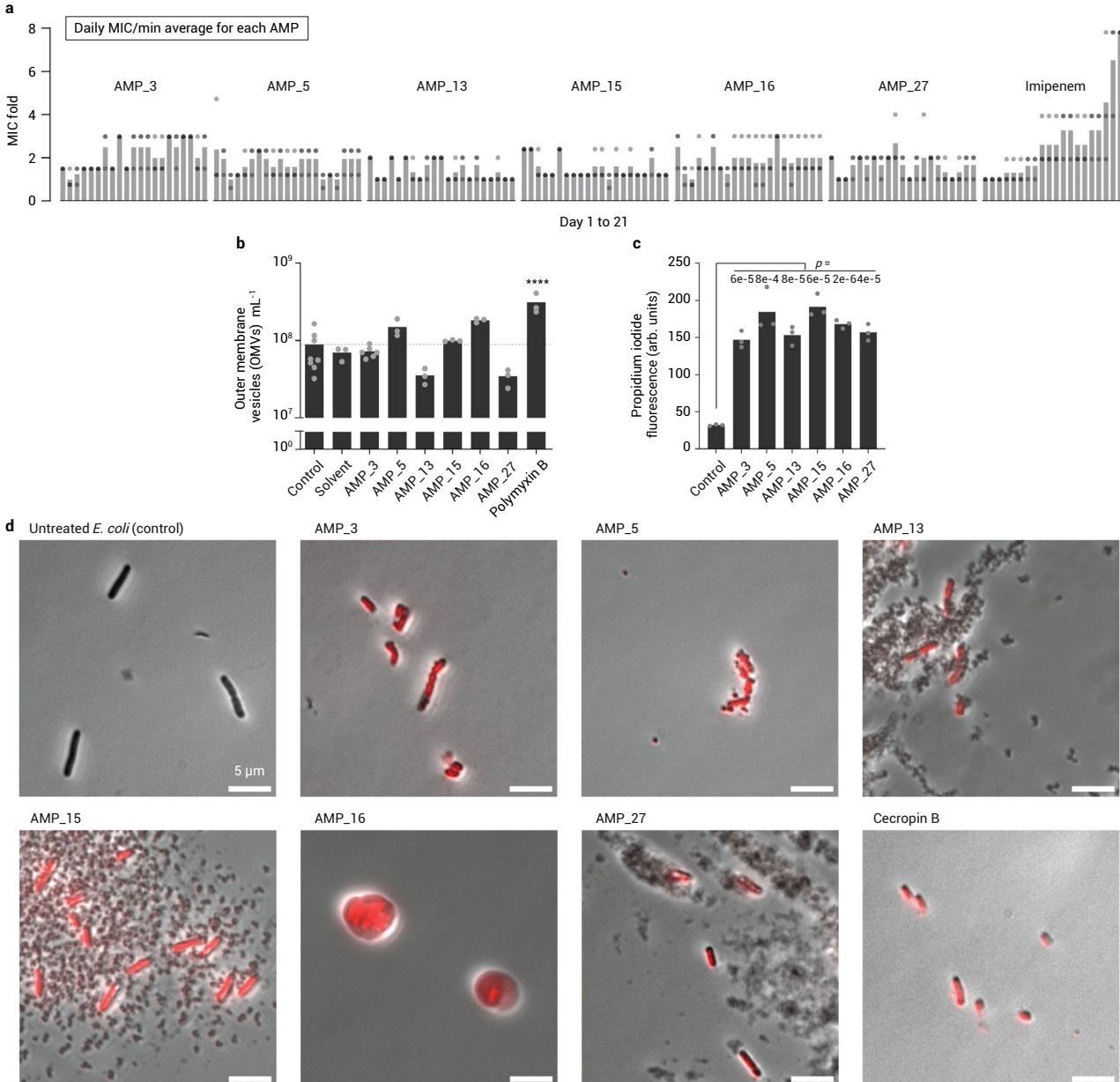

**Fig. 5 | Resistance and mode of action test of AMPs on *E. coli* and the mode of action assays. a** Twenty one days MIC measurement of daily culture passage each day from the cells grown in half-MIC concentrations of AMPs. The MIC values were normalized by the minimum of 21 daily averages for each AMP. Bars are the average of *n* = 3 independent experiments. See Supplementary Fig. 7 for raw MIC data. **b** Impact of different AMPs at the concentration of ¼ MIC on the number of *E. coli* outer membrane vesicles (OMVs) for untreated *E. coli* (control), (AMP) solvent, and six potent AMPs, and polymyxin B. Bars are the average of independent experiments *n* = 8 for the control, *n* = 6 for AMP#3, and *n* = 3 the rest. Statistics: ordinary one-way ANOVA *p* < 0.0001 (4.86×10⁻⁶), with Polymyxin B as the only significantly

different group. **c** Plate reader measured fluorescence (in arbitrary unit) of propidium iodide with *E. coli* cells untreated (control) or treated with AMPs. Bars are the average of *n* = 3 independent experiments. *P* values are for unpaired t-test of each AMP compared to untreated *E. coli* control. **d** Phase-contrast microscopy of *E. coli* cells untreated or treated with six potent AMPs and Cecropin B (a membrane disruptive AMP) and stained with propidium iodide (red). Source data for **a**–**c** and raw images in **d** (plus six more acquisitions for each, two images of *n* = 3 biological replicates with similar results) are provided as a Source Data file. Created with BioRender.com.

simple RNN regressor with an LSTM layer and two dense layers. The models were compiled with mean squared error loss and the Adam optimizer.

Gram-specific regressors. Due to structural differences in the cell membrane, we assumed there are differences between Gram-positive and Gram-negative bacteria in their response to AMPs. To capture these differences, we trained Gram-specific regressor models. We trained the Gram-negative model on 4619 unique AMP sequences and corresponding MIC measurements on *E. coli;* and the Gram-positive

model on 4089 AMP sequences and corresponding MIC measurements on *B. subtilis*. This approach improved the accuracy and efficiency of the regressor models (Supplementary Tables 1, 2, 4).

Training. We trained these models on pairs of data containing AMP sequences and their corresponding MIC measurements (in log 10). Based on a previous study[38], the nonAMP sequences were labeled to have a log MIC of 4. We interpreted the predicted MIC value as follows; below 3.5 as AMP, between 3.5–3.9 as potential AMP and above 3.9 as non-AMP. The regressor models were trained by the pooled AMP

and non-AMP data for 200 epochs. Model training metrics are provided in Supplementary Table 1.

## Sampling and prioritizing of AMPs

In each round of peptide synthesis, we selected thousands of random points from the VAEs latent space and from each reconstructed a peptide sequence. We omitted non-viable sequences and kept AMPs with 36-48 amino acids. This left us with viable peptide sequences (Supplementary Table 4). We then fed these sequences into the regressors and prioritized them based on the MIC prediction. 50-150 AMPs were chosen (either randomly among those predicted as AMP or from top-ranked sorted AMPs, Supplementary Tables 2, 5) for wet-lab experiments in five rounds making a list of over 500 AMPs tested in this work (Supplementary Table 2).

## Molecular dynamics (MD) simulations

All MD simulations were performed using Gromacs 2020.3[47] and the CHARMM36m forcefield[48] using an integration time step of 2 fs (partly 1 fs in the membrane equilibration; see Supplementary Table 12). Bonds including hydrogens were constrained using the LINCS algorithm[49]. Electrostatic interactions were computed using the Particle-Mesh Ewald (PME) algorithm[50] with a real-space cut-off for pairs further apart than 1.2 nm. Lennard-Jones interactions were smoothly switched to zero between 1 and 1.2 nm using the force-switch algorithm.

The protein, the membrane and the solvent (water and ions) were individually coupled to thermal baths set to 37 °C (310.15 K) using the v-rescale algorithm[51] with a time-constant of 1 ps. During the equilibration runs pressure coupling was handled by the Berendsen barostat[52], which was switched to the Parrinello-Rahman barostat[53] for all production runs. The barostat time-constant and compressibility factor were consistently set to 5 ps and $4.5 \times 10^{-5}$ bar$^{-1}$, respectively. Pressure coupling was applied semiisotropically (with the x and y dimensions coupled together) for systems with membrane and isotropically otherwise. The reference pressure was set to 1 bar. To counteract an energetic penalty for peptide insertion into the membrane due to finite size effects, we simulated each membrane system with three different lateral membrane tensions (0, 9, 17.1 bar). Therefore, the diagonal elements of the pressure tensor ($P_{XX}$, $P_{YY}$, $P_{ZZ}$) were set to

$$P_{XX} = P_{YY} = P - \frac{\Delta P}{3} \tag{1}$$

$$P_{ZZ} = P + 2\frac{\Delta P}{3} \tag{2}$$

with P as the reference pressure (1 bar) and $\Delta P$ as the desired lateral membrane tension. To ensure uncorrelated runs for the different tensions, the starting velocities of the atoms were randomly initialized according to the Maxwell-Boltzmann distribution. The MD simulations of the AMPs on membranes were performed for 1 μs each.

Visual analysis and renders used the VMD[54], PyMOL[55] and ChimeraX[56] software.

Membrane setup. Following earlier work and results from lipidomics experiments, we modeled membranes resembling the outer leaflet of the human plasma membrane (PM)[57,58] and the *E. coli* inner membrane (IM)[59]. The detailed compositions are summarized in Supplementary Tables 13, 14.

Using the CHARMM-GUI membrane builder[60,61], we generated 12.5 × 12.5 nm$^2$ patches of these model membranes, energy minimized them with a steepest descent algorithm until the largest force acting on any atom was below 1000 kJ mol$^{-1}$ and subsequently equilibrated them following the CHARMM-GUI equilibration scheme (summarized in Supplementary Table 12).

AMP system setup. Structures of the selected 30 AMPs and of Cecropin B were predicted using AlphaFold[29] while the short BP100 was modeled as a coil with initial angles φ = −60° and ψ = 30°, using the Molefacture Protein Builder plugin for VMD[54]. Since several of our AMPs have more than one cysteine (AMPs #2, #4, #6, #8, #10, #11, #12, #14, #18, #19), we next performed 1–2 μs long MD simulations of them surrounded by only water and ions (150 mM NaCl plus counter ions for overall neutralization) and with no disulfide bonds imposed. Based on the frequency and distance with which two cysteines in the respective structure interacted with each other in these simulations, we assigned a disulfide bond for AMP #2 (Cys11, Cys47) and disulfide bonds connecting the predicted β-sheets of AMP #8 (Cys29, Cys35) and AMP #14 (Cys10, Cys32). Due to the particularly high abundance of cysteine in AMP #2 and the potential structural bias from imposing one specific disulfide bond, we additionally simulated it without any imposed disulfide bonds.

These structures were then orientated so that their first principal axis was orthogonal to the z-axis and placed in proximity, but not yet bound to the equilibrated PM and IM. The box was subsequently solvated with TIP3P water[62] and NaCl ions were added to a concentration of 150 mM, ensuring overall neutrality by adding additional counter ions to the systems. The systems were energy minimized in the same way as the pure membrane systems and were then equilibrated for 5 ns. During minimization and equilibration, we applied position restraints on the peptide heavy atom with a force constant of 1000 kJ mol$^{-1}$ nm$^{-2}$.

## Cell-free production and activity test of AMPs

DNA fragments encoding AMPs were designed with T7 promoter (GAATTTAATACGACTCACTATAGGGAGA), RBS (TCTAGAGATTAAAG AGGAGAATACTAG) sequences upstream of the AMP coding region, and a T7 terminator (TACTCGAACCCCTAGCCCGCTCTTATCGGGCG GCTAGGGGTTTTTTGT) downstream. 500 DNA fragments were purchased from Twist Bioscience. A final concentration of 10 nM of each fragment was used for cell-free transcription and translation of AMPs using PUREfrex®2.0 kits (GeneFrontier #PF201-0.25-5-EX, purchased from Hölzel, Germany). In 384-well plates (BRAND, #781687), 30 μL volume of the cell-free reaction was made for each AMP and incubated for 4 h at 37 °C. The cell-free mix was directly used for the activity test on *E. coli* and *B. subtilis* or for the SDS-PAGEl of the functional AMPs.

*E. coli* MG1655 and *B. subtilis* PY79 were used as representatives of Gram-negative and Gram-positive bacteria. From LB agar plates into LB medium, three overnight cultures for each strain were made from three different colonies and grown while shaking at 37 °C. The next day, each was subcultured in LB (1:1000) and grown while shaking at 37 °C to OD ≈ 1. Cells were diluted in LB to 10$^4$ cfu mL$^{-1}$, and 16 μL of diluted cells were added to wells of a 384-well plate (Greiner Bio-One, #781185) in which 4 μL of the cell-free reaction mix (with AMPs produced) had been added beforehand. Cultures were mixed and the plate was sealed by a gas-permeable film (Carl Roth, #T093.1). OD$_{600}$ was measured every 10 min in a plate reader (Tecan Infinite® 200 PRO) shaking at 37 °C for 20 h. Growth curves were analyzed for AMPs impairing bacterial growth. We analyzed plates both by visual investigation of the microplates after 20 h as well as by visual analysis of the growth curves looking for stopped or slowed growth plotted OD$_{600}$ over time compared to the controls.

## SDS-PAGE of AMPs produced in the cell-free system

SDS-PAGE was used to detect produced functional AMPs in the cell-free reaction[23]. In brief, cell-free reactions were boiled for 3 min in 2x Tricine buffer (Bio-Rad, #1610739), loaded in 16.5% Mini-PROTEAN polyacrylamide Tris/Tricine gels (Bio-Rad, #4563065), and run for 5 h at 200 mA in the running buffer (10 mM Tris, 10 mM Tricine, and 0.01% SDS). Gels were then fixed for 1 h in 12% trichloroacetic acid and 1 h in 40% EtOH, 10% acetic acid, followed by overnight staining in QC

Colloidal Coomassie (Bio-Rad, #161-0803), 24 h of de-staining in water, and imaged using Intas GelStick Touch Imager.

## Measurement of minimum inhibitory concentration (MIC) and resistance test

Strains used for MIC measurements are *E. coli* MG1655, *B. subtilis* PY79, *E. faecium* (isolate from the gut of the cow), *S. aureus* DSM 11729, *K. pneumoniae* DSM 30104, *A. baumannii* (isolate from the human abdominal wall), *P. aeruginosa* DSM 1117, *Enterobacter* spp., *Y. pestis* EV76, *B. anthracis* Sterne, and *S. pneumoniae* D39. A commonly used standard protocol for determination of MIC for antimicrobials was used to measure the MIC of the AMPs taking into account all suggestions for cationic AMPs in the protocol[30]. Chemically synthesized peptides were dissolved in BSA (0.2% w/v) acetic acid (0.01% v/v) solution to have 10x of the highest concentration to be tested. In 96-well PCR plates (Axygen, #PCR-96-SG-C) two-fold serial dilutions were made from columns 1–10 in each row specified to each peptide and BSA (0.2% w/v) acetic acid (0.01% v/v) solution was pipetted into columns 11 and 12. Triplicates of 7.5 μL of each dilution were pipetted into polypropylene 96-well plates (Corning, #3359). Triplicates of bacterial overnight cultures in Mueller-Hinton broth 2 (MHB 2, Sigma-Aldrich, #90922) were prepared from three different colonies the day before and grown shaking at 37 °C, subcultured in the morning by diluting 1000x in MHB 2 grown shaking at 37 °C to OD ≈ 1. Bacterial cultures were then diluted with MHB 2 to $10^5$ cfu mL$^{-1}$ and 67.5 μL of each triplicate was added on top of peptides in columns 1–11 of 96-well plates. MHB 2 was added to column 12. The plates were sealed by adhesive films (VWR, #391-1262) and incubated at 37 °C for 20 h. MIC values were reported as the highest concentration of each AMP in which no visible growth was observed. For *S. pneumoniae* THY medium was used instead of MHB 2.

The same procedure was used for the resistance test except for cultures that from the 2nd day on, each of the triplicate cells grown in the highest AMP concentration (half MIC) was diluted 10,000x in MHB 2 and added to newly prepared peptides dilutions.

## Measurement of cytotoxicity (CC50)

Cytotoxicity assay was performed on HCT116 human colon cells (ATCC, #CCL-247™) using CellTiter 96® AQueous One Solution Cell Proliferation Assay (Promega, #G3580) which is a colorimetric method based on MTS (3-(4,5-dimethylthiazol-2-yl)−5-(3-carboxymethoxyphenyl)−2-(4-sulfophenyl)−2H-tetrazolium) for determining cell viability. The MTS tetrazolium compound (Owen's reagent) is reduced into a colored formazan product by NADPH or NADH produced by dehydrogenase enzymes in metabolically active cells.

On the 1st day, when cells reached the density of 50-80% of the covered surface, gently washed twice using 10 mL of DPBS (Gibco, #14190367). 1 mL trypsin (Capricorn, #TRY-1B) was added, 5 min incubated at 37 °C and 9 mL medium was added including DMEM high glucose (Capricorn, #DMEM-HPA), 10% v/v fetal bovine serum (FBS, Capricorn, #FBS-11A), and antibiotic mix for cell culture (Capricorn, #PS-B). The culture was transferred into a 15 mL falcon, and spun down at 1000 rpm for 3 min. The supernatant was sucked out and 10 mL of fresh media was added and transferred. Cells were diluted by the medium to have 5000 cells in 36 μL. 36 μL of the cell culture was pipetted into wells of a 384-well plate (Greiner, #781185). The last well received only media. Cells were incubated at 37 °C, 5% CO$_2$, for 24 h. On the 2nd day, peptides were prepared in two-fold serial dilutions starting from a final concentration of 250 μM. Columns 11 and 12 received only water. 4 μL of each peptide dilution was added to wells of the cell culture plate prepared on the 1st day and the plate was put at 37 °C, 5% CO$_2$, for 24 h. On the 3rd day, 8 μL of CellTiter 96® AQueous One Solution and 10 μL SDS 10% were added to each well and after 90 min incubation at 37 °C, absorbance at 490 nm was measured and corrected by the value of wells with only medium. CC50 values, the

concentration of each AMP killing 50% of cells, were calculated using Graphpad Prism 9.

## Measurement of hemolytic activity (HC50)

Human blood was washed three times with PBS and resuspended in 2 V PBS. AMPs with an initial concentration of 250 μM were titrated in 96-well polypropylene plates (V-bottom, Greiner Bio-One GmbH). 5 μL AMP dilutions were overlaid with 45 μL of washed and concentrated human erythrocytes, and the plates were sealed and incubated at 37 °C for 1 h. 40 μL of supernatant were transferred after final centrifugation at 1000 x g for 5 min at room temperature to ELISA plates and absorbance was measured at 405 nm. Triton X 100 treated erythrocytes served as positive control. HC50 values, the concentration of each AMP lysing 50% of RBCs, were calculated using GraphPad Prism.

## Mode of action assay and microscopy using propidium iodide (PI)

Plate reader assay. Three colonies were picked to culture *E. coli* MG1655 cells in LB at 37 °C to the exponential phase. Cells were harvested by centrifugation at 4000 x g and washed three times in 10 mM PBS (pH = 7.0), and adjusted to OD$_{600}$ = 1 with 10 mM PBS. 10 μL of the cells in PBS were mixed with 10 μL of AMPs to a final concentration of 4×MIC and incubated at 37 °C for 1 h. 20 μM final concentration of PI[63] was added to each of the AMP-treated and untreated samples and incubated at 37 °C for 30 min in the dark. Fluorescence was recorded at an excitation of 535 nm and emission of 615 nm was measured using a Tecan Infinite® 200 PRO plate reader.

Sample preparation for microscopy. *E. coli* was grown in LB at 37 °C to the exponential phase and diluted to $10^8$ cfu mL$^{-1}$ in fresh LB. 50 μL of diluted cells were pipetted in a 1.5 mL tube and 50 μL of AMPs was added at 4×MIC final concentration together with 20 μM PI. The mixture was incubated at 37 °C for 1 h while shaking. 1 μL of samples was loaded onto agarose pads (2% agarose in PBS) and imaged using a Zeiss AxioPlan 2 upright widefield microscope equipped with a 100x NeoFluor phase contrast objective and a FluoArc HBO lamp. Fluorescence of PI was recorded using TxRed HC Filter set (AHF, F36-504).

Image processing. The phase contrast and fluorescence images were overlaid. The dynamic range of the fluorescence channel was set to a minimum of 125 AU to remove background fluorescence, while the contrast of the phase contrast channel was manually adjusted for better visualization. For comparison of the different phenotypes, 232 pixels x 232 pixels regions of interest were cropped. Raw image files are provided in this study.

## Outer membrane vesicle (OMV) release of *E. coli* after AMP treatment

*Escherichia coli* MG1655 was grown on MacConkey agar (Carl Roth, Karlsruhe, Germany) plates overnight. For overnight culture, a single colony was used for inoculation of 2 mL LB media at 37 °C, 160 rpm (MaxQ 6000, Thermo Fisher Scientific, Karlsruhe, Germany). Bacterial culture was transferred to 10 mL fresh LB media and incubated (1 h, 37 °C, 160 rpm). The required amount of bacteria was treated with ¼ MIC of the AMPs #3, #5, #10, #13, #15, #16, and #27, left untreated for control or was treated with acetic acid and BSA as solvent control (90 min, 37 °C, 160 rpm). Samples were centrifuged (4,500 x g, 15 min, 4 °C; Multifuge X3R, Thermo Fisher Scientific), the supernatant was sterile filtered (0.22 μm) and afterwards concentrated by the factor of 20 using 100 kDa molecular weight cut-off filters (Merck KGaA, Darmstadt, Germany). To determine the number of released vesicles, samples were measured by nano-flow cytometry (nFCM) using a NanoAnalyzer (NanoFCM Co., Ltd, Nottingham, UK)[64].

## Chemical peptide synthesis of AMPs

Materials. All commercially available reagents were purchased from the following companies, and used without further purification:

thioanisol (#T28002), 1,2-ethandithiol (EDT, #8.00795), N-Methyl-2-pyrrolidone (NMP, #M79603) from Sigma Aldrich (USA); piperidine (#T6146), 2 mL polypropylene reactors with plunger and frit pore size 25 μm (#7926.1) from Carl Roth (Germany); 2,6-lutidine (#10731354), palladium acetate (#441390010), phenylsilane, trifluoroacetic acid (TFA, #293812500) from Acros (USA); microscale columns from Intavis (#35.091) (Germany); Fmoc-Gly-OH (#FAA1050.0100), Fmoc-L-Asn(Trt)-OH (#FAA1015.0100), Fmoc-L-Asp(tBu)-OH (#FAA1356.0100), Fmoc-L-His(Trt)-OH (#FAA1090.0100), Fmoc-L-Ile-OH (#FAA1110.0100), Fmoc-L-Met-OH (#FAA1150.0100), Fmoc-L-Phe-OH (#FAA1175-0100), Fmoc-L-Pro-OH*H2O (#FAA1185.0100), Fmoc-L-Thr(tBu)-OH (#FAA1210.0100), Fmoc-L-Tyr(tBu)-OH (#FAA1230.0100), Fmoc-L-Val-OH (#FAA1245.0100), TentaGel S RAM resin (#S-30023) from Iris Biotech (Germany); Fmoc-L-Ala-OH-OH (#05001), Fmoc-L-Arg(Pbf)-OH (#CC05005), Fmoc-L-Cys(Trt)-OH (#CC05009), Fmoc-L-Gln(Trt)-OH (#CC05012), Fmoc-L-Glu(OtBu)-OH (#CC05013), Fmoc-L-Leu-OH (#CC05017), Fmoc-L-Lys(Boc)-OH (#CC05018), Fmoc-L-Ser(tBu)-OH (#CC05023), Fmoc-L-Trp(Boc)-OH (#CC05076), N,N'-diisopropylcarbodiimid (DIC, #CC01002), and Oxyma (#CC01024) from Carbolution (Germany); acetic anhydride (Ac2O) from Grüssing GmbH (Germany); peptide grade dimethylformamide (DMF, #1003976025) and HPLC grade acetonitrile (MeCN) (#34851-2.5 L) from Merck (Germany); Ultrapure water of type 1 was obtained with a MicroPure Water Purification System from TKA (Germany).

Solid-phase peptide synthesis. All peptides were synthesized via the Fmoc-solid phase strategy. The synthesis was carried out by an automated peptide synthesizer using INTAVIS ResPep SLi instrument for a 5 μmol scale. Higher amounts of peptides were achieved by running multiple 5 μmol syntheses in parallel. For all peptides, the TentaGel S RAM (0.22 mmol/g) resin was used.

Automated solid-phase peptide synthesis (INTAVIS ResPep SLi):
- The conditions, reagents, and corresponding volumes of this synthesis protocol correspond to a 5 μmol scale synthesis. No mixing was performed during incubation or reaction time. In the coupling step, the temperature was set to 40 °C.
- Swelling: The appropriate amount of resin (23 mg) was swelling in 200 μL DMF for 30 min.
- Deprotection of temporal Fmoc protecting groups: Piperidine (150 μL, 20% in DMF) was added to the resin and incubated for 5 min. This step was repeated, and the resin was filtered off, and washed with DMF (1 × 300 μL, 3 × 225 μL).
- Coupling of amino acids: In a mixing vial the machine automatically added 53 μL of the corresponding Fmoc-amino acid (4 eq, 0.5 M), 15 μL Oxyma (4 eq, 2 M), 13 μL DIC (4 eq, 2 M) each in DMF and 29 μL NMP. The resulting solution was activated by waiting for 1 min before addition to the resin. This suspension was incubated for 15 min. Next, the resin was filtered off, and the coupling was repeated. No washing was performed after the coupling.
- Capping: 150 μL of a lutidine/Ac2O/DMF 6:5:89 solution was added to the resin and incubated for 8 min. The resin was filtered off and washed with DMF (3 × 225 μL).
- After the last coupling, the resin was washed with DMF (1 ×300 μL, 3 × 225 μL), ethanol (4 × 150 μL) and CH2Cl2 (5 × 150 μL). The resin was finally dried under continuous air flow for 5 min.

Final Cleavage, Purification, and Characterization:
- Cleavage and deprotection of the amino acid side chains: All peptides have been cleaved from a dry resin previously washed with 5x DMF and 10x CH2Cl2 following the last step of the synthesis protocol. Depending on the total number of Cys, Met, or Trp, one of the cleavage cocktails in Supplementary Table 15 was utilized. Cleavage cocktail A was used as the initial test cleavage after complete synthesis[65]. In the cases where oxidation was observed after cleavage, cocktail B was applied[66,67]. For 5 μmol of resin, 2 mL of cleavage cocktail was prepared. The total volume was increased by 1.5-fold for peptides containing more than eight arginine residues. The dry resin was loaded into 2 mL reactors with a plunger and the frit was treated with the corresponding mixture and shaken for 2.5 h and filtered off. The resin was washed with 1 mL of TFA and the filtrates were combined. The TFA content of the filtrate was reduced via a gentle nitrogen flow. Next, ice-cold diethyl ether (DEE) (1.00 mL of DEE for 100 μL cocktail) was added to precipitate the final peptide. The precipitated peptide was centrifuged (8000 rpm, 4 °C, 5 min), the supernatant was discarded and the pellet washed once more; i.e. redissolved, and precipitated with cold DEE. Afterward, the peptide pellet was dissolved in ultrapure water/MeCN (70:30) with 0.1% of TFA (more MeCN was added when insoluble, not exceeding 1:1) to be purified.
- Purification: The peptides were purified by reverse-phase (RP)-HPLC using a preparative Agilent 1260 Infinity II Series HPLC-system (Agilent Technologies) with column 1 (Supplementary Table 16). An isocratic regime during the first five minutes for column equilibration, followed by the respectively stated linear gradient in 25 min (gradient is specified at the respective peptide). The detection was carried out by measuring absorption at the wavelengths: 220 nm and 260 nm. Ultrapure water (A) and MeCN (B) were employed as eluents with an addition of 0.1% of TFA in both solvents.
- Characterization: The freeze-dried products were identified via analytical HPLC-MS on an Agilent 1260 Infinity II Series HPLC-system (Agilent Technologies) using column 2 (Supplementary Table 16). The detection was carried out by measuring absorption at the wavelengths: 220 nm and 260 nm. Ultrapure water (A) with an addition of 0.05% of TFA and MeCN (B) addition of 0.03% of TFA were employed as eluents. HR-ESI-MS was performed for identification on an LTQ-FT Ultra device (Thermo Fischer Scientific). HPLC chromatogram of purified peptides are provided in Supplementary Figs. 8–34.

### Data collection/analysis tools
Bacterial growth data, PI fluorescence, and absorbance values were collected on Tecan Infinite 200 Pro plate reader with Magellan™ standard software. Outer membrane vesicles were collected using NanoAnalyzer (NanoFCM Co., Ltd, Nottingham, UK) with NanoFCM software (NF Profession V1.08). Images were collected using with a Zeiss AxioPlan 2 upright widefield microscope with MetaMorph version 6.2r6 software. Data was analyzed using Python 3 scripts, MS Excel 2021, GraphPad Prism v9 and FiJi version 1.54 f.

### Reporting summary
Further information on research design is available in the Nature Portfolio Reporting Summary linked to this article.

## Data availability
Data that are necessary to interpret, verify and extend the research in the article, have been made available to readers. The training data used in this study along with the sequence of 500 tested AMPs, data underlying Supplementary Figs. 4 and 5 generated in this study can be found at https://github.com/amirpandi/Deep_AMP. Sequence of the 30 functional AMPs and source data underlying Fig. 4a generated in this study are provided in Supplementary Information (Supplementary Tables 5 and 10, respectively). The source data underlying Fig. 2b, Fig. 5a-d, Supplementary Fig. 1, Supplementary Fig. 2, and Supplementary Fig. 7 generated in this study are provided as Source Data files. Source data are provided with this paper.

## Code availability

All deep learning models were built, trained, and tested using Keras 1.0 with TensorFlow 2.0 backend using Python 3.9 in the Google Colab pro environment. The deep learning codes and models developed in this study can be found at https://github.com/amirpandi/Deep_AMP. MD simulation setups can be found at https://doi.org/10.5281/zenodo.7327525.

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

## Acknowledgements

This work was supported by a European Molecular Biology Organization (EMBO) long-term postdoctoral fellowship to A.P. (ALTG 165-2020), the Max Planck Society and the Gordon and Betty Moore Foundation (https://doi.org/10.37807/GBMF10652) to T.J.E, German Research Council DFG; AB 792/1-1 (423428279) to F.A., German Research Council TRR81: 'Chromatin Changes in Differentiation and Malignancies' TRR81/3, Z04 (109546710) to V.T.T. and O.V., the Bundesministerium für Bildung und Forschung (Federal Ministry of Education and Research, PermedCOPD – FKZ 01EK2203A; ERACoSysMed2—SysMed-COPD—FKZ 031L0140), the Deutsche Forschungsgemeinschaft (SFB/TR-84 TP C01) to B.S., the Hessisches Ministerium für Wissenschaft und Kunst (LOEWE Diffusible Signals LOEWE-Schwerpunkt Diffusible Signals) to A.L.J. and B.S. S.L.S. and G.H. acknowledge financial support by the Clusterprojekt ENABLE funded by the Hessian Ministry for Science and the Arts, and the Collaborative Research Center 1507 funded by the Deutsche Forschungsgemeinschaft (DFG, German Research Foundation), and thank the Max Planck Computing and Data Facility (MPCDF) for computational resources. M.K. acknowledges funding support from INRAe's MICA department, Université Paris-Saclay, Ile-de-France (IdF) region's DIM-RFSI, ANR DREAMY (ANR-21-CE48-003, and ANR iCFree grant (ANR-20-BiopNSE)). We thank R. Weiss, P. Wichmann, A.M. Küffner, S. Scholz, R. Inckemann, C. Diehl, A. Yazdizadeh Kharrazi, J. Zarzycki, and H. He for technical and experimental support, as well as fruitful discussion. In Fig. 1d, the PDB structures 1MSW[68] and 5IT8[69] were used and edited with Biorender.com. Figures were created with Biorender.com.

## Author contributions

T.J.E. and A.P. conceived the study and wrote the manuscript with contributions from other authors, as indicated in the following. A.Z. developed deep learning codes and simulations and wrote the corresponding method section and performed BLAST analysis. A.P. and Y.F. performed cell-free protein synthesis and the initial bioactivity tests. M.K. performed the RBS calculator and kinfold calculations. S.L.S. performed and, together with G.H. analyzed molecular dynamics simulation and wrote the corresponding methods and results section. A.P., D.A., B.K., P.B. and H.v.B. performed and analyzed MIC, hemolysis, and cytotoxicity tests. V.T.T. performed chemical peptide synthesis and analysis via automated solid-phase and wrote the corresponding methods section. A.P., E.B., and C.S. performed and analyzed the mode of action assays. M.B. performed and analyzed the outer membrane vesicle experiments and wrote the corresponding results and methods sections. C.P. established the OMV quantification and performed the OMV quantification in the Core Facility for Extracellular Vesicles. E.P.v.s., H.B.B., H.v.B., W.B., A.L.J., F.A., B.S., G.H., O.V., and T.J.E. supervised the work. All authors provided input on the manuscript and confirmed the final draft.

## Funding

## Competing interests

The authors declare no competing interests.

## Additional information

[1]Department of Biochemistry and Synthetic Metabolism, Max Planck Institute for Terrestrial Microbiology, Marburg, Germany. [2]Bundeswehr Institute of Microbiology, Munich, Germany. [3]Department of Chemistry, Philipps-University Marburg, Marburg, Germany. [4]Department of Theoretical Biophysics, Max Planck Institute of Biophysics, Frankfurt am Main, Germany. [5]Institute for Lung Research, Universities of Giessen and Marburg Lung Center, Philipps-University Marburg, German Center for Lung Research (DZL), Marburg, Germany. [6]Université Paris-Saclay, INRAe, AgroParisTech, Micalis Institute, Jouy-en-Josas, France. [7]German Center for Infection Research (DZIF), Munich, Germany. [8]Fraunhofer Institute for Translational Medicine and Pharmacology (ITMP), Immunology, Infection and Pandemic Research, Munich, Germany. [9]Department of Natural Products in Organismic Interactions, Max Planck Institute for Terrestrial Microbiology, Marburg, Germany. [10]Institute for Tumor Immunology, Center for Tumor Biology and Immunology, Philipps-University Marburg, Marburg, Germany. [11]Core Facility Extracellular Vesicles, Center for Tumor Biology and Immunology, Philipps-University of Marburg, Marburg, Germany. [12]Molecular Biotechnology, Department of Biosciences, Goethe University Frankfurt, Frankfurt am Main, Germany. [13]Department of Chemistry, Chemical Biology, Philipps-University Marburg, Marburg, Germany. [14]Senckenberg Gesellschaft für Naturforschung, Frankfurt, Germany. [15]SYNMIKRO Center of Synthetic Microbiology, Marburg, Germany. [16]Core Facility Flow Cytometry – Bacterial Vesicles, Philipps-University Marburg, Marburg, Germany. [17]Department of Medicine, Pulmonary and Critical Care Medicine, University Medical Center Marburg, Philipps-University Marburg, Marburg, Germany. [18]Institute for Lung Health (ILH), Giessen, Germany. [19]Member of the German Center for Infectious Disease Research (DZIF), Marburg, Germany. [20]Institute for Biophysics, Goethe University Frankfurt, Frankfurt am Main, Germany. ✉e-mail: amir.pandi@mpi-marburg.mpg.de; toerb@mpi-marburg.mpg.de

