## [Peer Review File · Nature Communications]

Cell-free biosynthesis combined with deep learning accelerates de novo-development of antimicrobial peptidesEditorial Note: This manuscript has been previously reviewed at another journal that is not operating a transparent peer review scheme. This document only contains reviewer comments and rebuttal letters for versions considered at *Nature Communications* .

REVIEWERS' COMMENTS

Reviewer #1 (Remarks to the Author):

This paper by Pandi et al. describes a deep learning approach consisting of a VAE to generate new peptides, coupled with a regressor to predict their antimicrobial activity. Lead peptide candidates were then synthesized for in vitro validation using cell-free synthesis.

In summary, although the in vitro experimental effort is commendable, the computational approach lacks novelty as it is similar to other reports in the literature. Moreover, the lack of in vivo validation and the uncertain purity of the peptides generated lessened my enthusiasm. Additional experiments would need to be performed to address these issues.

Reviewer #2 (Remarks to the Author):

the authors have addressed my concerns and I support publication based on the novel workflow.

Reviewer #3 (Remarks to the Author):

The comments by me and the other reviewers have been taken very seriously. I agree with the authors that the combination of the computational approach with the automated peptide production workflow is exciting and novel. We are happy with the textual and scientific adjustments that have been made to the paper based on our comments.

Reviewer #1 (Remarks to the Author):

This paper by Pandi et al. describes a deep learning approach consisting of a VAE to generate new peptides, coupled with a regressor to predict their antimicrobial activity. Lead peptide candidates were then synthesized for in vitro validation using cell-free synthesis.

In summary, although the in vitro experimental effort is commendable, the computational approach lacks novelty as it is similar to other reports in the literature. Moreover, the lack of in vivo validation and the uncertain purity of the peptides generated lessened my enthusiasm. Additional experiments would need to be performed to address these issues.

Response:

As mentioned throughout the previous revision and responses to the reviewer's comments, the novelty of our work lies in the combination of AMP screening via cell-free protein synthesis and de novo AMP design.

Regarding the comment for in vivo validation, please refer to our response in the previous round. In line with the reviewer's comment, we modified our claims throughout the manuscript listed below.

Action:

We added the term "in vitro" in the following sections:

- Results subheadings in Lines 249 and 262: "De novo AMPs show broad-band activities **in vitro**" and "No resistance was developed against de novo AMPs **in vitro**"
- Discussion, Line 298: "In this work, we describe the design and validation of 30 de novo AMPs, of which six show broad-band activity **in vitro**."

For further clarification, we made the following modification to the Discussion section in Lines 326-327: "The resulting AMPs have several features that (after in vivo validation) could contribute to their successful translation into therapeutic applications,"

Reviewer #2 (Remarks to the Author):

the authors have addressed my concerns and I support publication based on the novel workflow.

Response:

We are pleased to hear that the reviewer's concerns have been addressed, and they support publication of our work.

Reviewer #3 (Remarks to the Author):

The comments by me and the other reviewers have been taken very seriously. I agree with the authors that the combination of the computational approach with the automated peptide production workflow is exciting and novel.

We are happy with the textual and scientific adjustments that have been made to the paper based on our comments.

Response:

We are glad that the reviewer found our revision satisfactory and are thankful for their endorsement of the novelty of our combined approach.